# KRAP tethers IP$_3$ receptors to actin and licenses them to evoke cytosolic Ca$^{2+}$ signals

Nagendra Babu Thillaiappan [1,2✉], Holly A. Smith[1], Peace Atakpa-Adaji[1] & Colin W. Taylor [1✉]

Regulation of IP$_3$ receptors (IP$_3$Rs) by IP$_3$ and Ca$^{2+}$ allows regenerative Ca$^{2+}$ signals, the smallest being Ca$^{2+}$ puffs, which arise from coordinated openings of a few clustered IP$_3$Rs. Cells express thousands of mostly mobile IP$_3$Rs, yet Ca$^{2+}$ puffs occur at a few immobile IP$_3$R clusters. By imaging cells with endogenous IP$_3$Rs tagged with EGFP, we show that KRas-induced actin-interacting protein (KRAP) tethers IP$_3$Rs to actin beneath the plasma membrane. Loss of KRAP abolishes Ca$^{2+}$ puffs and the global increases in cytosolic Ca$^{2+}$ concentration evoked by more intense stimulation. Over-expressing KRAP immobilizes additional IP$_3$R clusters and results in more Ca$^{2+}$ puffs and larger global Ca$^{2+}$ signals. Endogenous KRAP determines which IP$_3$Rs will respond: it tethers IP$_3$R clusters to actin alongside sites where store-operated Ca$^{2+}$ entry occurs, licenses IP$_3$Rs to evoke Ca$^{2+}$ puffs and global cytosolic Ca$^{2+}$ signals, implicates the actin cytoskeleton in IP$_3$R regulation and may allow local activation of Ca$^{2+}$ entry.

[1] Department of Pharmacology, Tennis Court Road, Cambridge, UK. [2] Department of Basic Medical Sciences, College of Medicine, QU Health, Qatar University, Doha, Qatar. ✉email: nagendrababu@qu.edu.qa; cwt1000@cam.ac.uk

Cytosolic $Ca^{2+}$ signals regulate diverse activities in all eukaryotic cells[1]. Most of these signals are due to the opening of $Ca^{2+}$-permeable channels, which then allow $Ca^{2+}$ to flow into the cytosol across either the plasma membrane (PM) or the membranes of intracellular organelles, primarily the endoplasmic reticulum (ER). Inositol 1,4,5-trisphosphate receptors (IP$_3$Rs), the most widely expressed of these $Ca^{2+}$ channels, reside in the membranes of the ER from which they release $Ca^{2+}$ when they bind IP$_3$[2,3]. This $Ca^{2+}$ flux delivers $Ca^{2+}$ to the cytosol and to the cytosolic surface of other organelles, notably mitochondria[4] and lysosomes[5]. By depleting the ER of $Ca^{2+}$, IP$_3$-evoked $Ca^{2+}$ release also stimulates store-operated $Ca^{2+}$ entry (SOCE) across the PM. Stromal interaction molecule 1 (STIM1), which straddles the ER membrane, detects the loss of ER $Ca^{2+}$ through its luminal $Ca^{2+}$-binding sites causing it to unfurl cytosolic domains. These domains reach across narrow membrane contact sites (MCS) between the ER and PM to contact Orai $Ca^{2+}$ channels, causing them to open and allow $Ca^{2+}$ to flow into the cell through the SOCE pathway[6,7]. IP$_3$Rs are therefore $Ca^{2+}$-signalling hubs: in all animal cells, they link the extracellular stimuli that evoke the formation of IP$_3$ to delivery of $Ca^{2+}$ from the ER to the cytosol or other organelles, and to the regulation of $Ca^{2+}$ entry across the PM through SOCE.

IP$_3$ is not alone sufficient to stimulate the opening of an IP$_3$R. Instead, IP$_3$ binding to all four subunits of a tetrameric IP$_3$R[8] primes it to respond to $Ca^{2+}$, which then evokes channel opening[3]. In the presence of IP$_3$, IP$_3$Rs can thereby evoke regenerative signals through $Ca^{2+}$-induced $Ca^{2+}$ release (CICR). The smallest of these regenerative events are $Ca^{2+}$ puffs, which are local cytosolic $Ca^{2+}$ signals that arise from coordinated openings of a few clustered IP$_3$Rs as $Ca^{2+}$ released by an open IP$_3$R stimulates the opening of its neighbours[9] (Supplementary Fig. 1a). $Ca^{2+}$ puffs, which can be evoked by all three IP$_3$R subtypes[10,11], allow local $Ca^{2+}$ signalling and they have been thought to be the building blocks for larger $Ca^{2+}$ signals, although that view has recently been challenged[12]. Cells typically express several thousand IP$_3$Rs, most of which are mobile[13–15], yet $Ca^{2+}$ puffs, whether evoked by endogenous signalling pathways or by the uniform release of cytosolic IP$_3$ from a caged precursor, occur repeatedly at rather few immobile sites within a cell[16–21]. In seeking to address this long-standing conundrum we recently reported that $Ca^{2+}$ puffs occur at only a few immobile IP$_3$R clusters alongside the PM and adjacent to the sites where STIM1 accumulates when SOCE is activated[20]. We described these responsive IP$_3$R clusters as licensed IP$_3$Rs to suggest an additional level of regulation that precedes gating by IP$_3$ and $Ca^{2+}$.

Here, we identify this additional form of regulation and demonstrate that IP$_3$Rs are licensed to respond when they are tethered to actin by KRas-induced actin-interacting protein (KRAP). We show that all $Ca^{2+}$ signals, whether local or global, require KRAP-mediated licensing of IP$_3$Rs and that endogenous KRAP limits the capacity of IP$_3$Rs to respond. This additional, and obligatory, level of IP$_3$R regulation reveals an important role for the actin cytoskeleton in regulating IP$_3$-evoked $Ca^{2+}$ signals and suggests mechanisms whereby local depletion of the ER might control SOCE.

## Results

### Licensed IP$_3$Rs associate with actin.
We used HeLa cells with endogenous type 1 IP$_3$Rs tagged with enhanced green fluorescent protein (EGFP) to address the mechanisms that license immobile IP$_3$Rs adjacent to the PM to evoke $Ca^{2+}$ puffs[20]. IP$_3$R1 is the major subtype in HeLa cells[22], but we showed previously that an antibody to EGFP immunoprecipitated all three IP$_3$R subtypes

from EGFP-IP$_3$R1 HeLa cells in the same ratio as their overall expression, confirming that EGFP-IP$_3$R1 subunits assemble with the other IP$_3$R subtypes[20]. Thus, EGFP-IP$_3$R1 puncta probably report the presence of all three IP$_3$R subtypes in EGFP-IP$_3$R1 HeLa cells.

We first considered whether $Ca^{2+}$ leaking across the PM might provide a local increase in cytosolic free $Ca^{2+}$ concentration ($[Ca^{2+}]_c$) that then selectively stimulates IP$_3$Rs immediately beneath the PM. Previous work with SH-SY5Y neuroblastoma cells[9] and human embryonic kidney (HEK) cells[11] established that $Ca^{2+}$ puffs do not require extracellular $Ca^{2+}$. We confirmed these findings in HeLa cells by demonstrating that the frequency and amplitude of $Ca^{2+}$ puffs evoked by photolysis of caged IP$_3$ (ci-IP$_3$) were unaffected by removing extracellular $Ca^{2+}$ (Fig. 1a, b). We conclude that licensing cannot be due to sensitization of IP$_3$Rs near the PM by $Ca^{2+}$ leaking across the PM.

The immobility of licensed IP$_3$Rs alongside the PM[20], the presence of a cortical actin cytoskeleton in most animal cells[23], evidence that IP$_3$Rs interact with actin[24–26], and identification of actin-binding proteins as partners of IP$_3$Rs in HeLa cells[27] led us to consider whether actin might anchor licensed IP$_3$Rs. Total internal reflection fluorescence microscopy (TIRFM) revealed that $27 \pm 7\%$ of EGFP-IP$_3$R1 clusters colocalize with actin filaments (Fig. 1c, Supplementary Movie 1); this fraction is similar to the immobile fraction of peripheral IP$_3$Rs (~25%) reported previously[20]. Live-cell imaging confirmed that immobile IP$_3$R puncta selectively associate with actin (Fig. 1d, Supplementary Fig. 1b–e). Cytochalasin D and latrunculin A caused actin depolymerization, and as actin filaments retracted many immobile IP$_3$R puncta retreated with them and remained associated with residual filaments (Supplementary Fig. 1f–i, Supplementary Movies 2, 3)[28]. We detected no association of IP$_3$Rs with intermediate filaments[29–32] (Supplementary Fig. 2, Supplementary Movie 4). These results suggest that IP$_3$R puncta are immobilized alongside the PM by association with actin.

### KRas-induced actin-interacting protein ties IP$_3$Rs to actin.
KRas-induced actin-interacting protein (KRAP), now designated IP$_3$ receptor-interacting domain-containing protein 2 (ITPRID 2), was originally identified as a large actin-binding protein that is over-expressed in a colon cancer cell line expressing activated KRas[33]. KRAP is now known to be widely expressed[33,34] and to associate with all three IP$_3$R subtypes[29,35,36]. Results from co-immunoprecipitation and immunocytochemical analyses suggest that the N-terminal region of KRAP may interact with IP$_3$Rs, while the C-terminal may interact with actin (Supplementary Fig. 3a); it remains unclear whether these interactions are direct or via intermediary proteins[33,35,37,38]. The functions of KRAP are poorly understood, but it affects both the subcellular distribution of IP$_3$Rs[29] and IP$_3$-evoked $Ca^{2+}$ release[35,39]. Proteins related to KRAP are also implicated in $Ca^{2+}$ signalling[33,40–43] (Supplementary Fig. 3a). These observations, alongside evidence from the HeLa cell interactome suggesting that KRAP and IP$_3$Rs share partners, including several that interact with actin[27] (Supplementary Fig. 3b), prompted us to consider whether KRAP might license IP$_3$Rs.

HeLa cells express KRAP[35] (Supplementary Fig. 3c), and TIRFM and spinning-disc confocal microscopy revealed that endogenous KRAP form puncta, some of which colocalize with a subset of EGFP-IP$_3$Rs (Fig. 1e). The colocalization was restricted to regions close to the PM (Fig. 1f, g, Supplementary Fig. 3d). We used an object-based colocalization method[44] to study nearest-neighbour distances between IP$_3$R and KRAP puncta. This

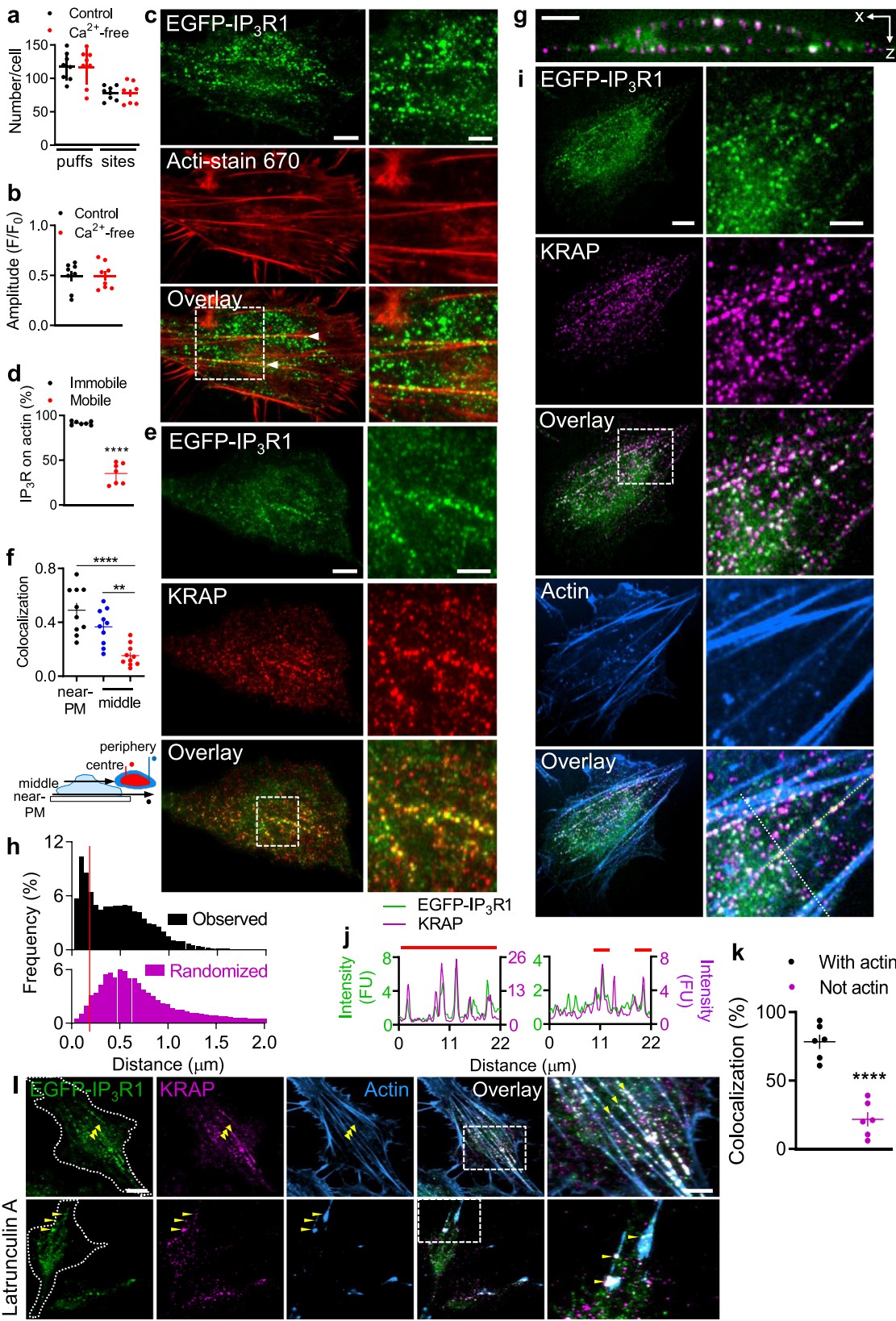

confirmed that $28 \pm 7\%$ of IP$_3$R puncta colocalized with KRAP, significantly more than expected from randomly distributed puncta (Fig. 1h, Supplementary Fig. 3e). Most IP$_3$R puncta that colocalized with KRAP were associated with actin filaments (Fig. 1i–k), but only with filaments close to the PM (Supplementary Fig. 3f, Supplementary Movies 5, 6). Furthermore,

colocalized IP$_3$R-KRAP puncta retreated with residual actin as it depolymerized after the addition of latrunculin A (Fig. 1l).

Immobile IP$_3$R puncta are stable, but loose, confederations of about eight IP$_3$Rs, some of which are too far apart to interact directly (Fig. 2a–d)[20]. Using super-resolution microscopy (stochastic optical reconstruction microscopy, STORM) and TIRFM,

**Fig. 1 Immobile IP$_3$Rs associate with actin and KRAP. a** Numbers of Ca$^{2+}$ puffs and Ca$^{2+}$-release sites detected in EGFP-IP$_3$R1 HeLa cells in the 20 s after photolysis of ci-IP$_3$ with or without extracellular Ca$^{2+}$ (2.5 mM BAPTA added immediately before UV flash). Mean ± s.d., 8 cells. **b** Mean puff amplitudes for individual cells (mean ± s.e.m., $n = 8$ cells). The results establish that Ca$^{2+}$ leaking across the PM does not account for Ca$^{2+}$ puffs occurring near the PM. **c** TIRF images of EGFP-IP$_3$R1 HeLa cells show some IP$_3$R puncta colocalized with actin filaments (arrows). Manders split coefficient for IP$_3$Rs: 0.27 ± 0.07 (mean ± s.d., 11 cells from 5 independent dishes). Scale bars (c,e,i,l), 10 μm (5 μm in enlargements). **d** IP$_3$R puncta categorized as mobile or immobile and whether they colocalize with actin filaments (mean ± s.d., 7 cells). $^{****}P < 0.0001$, Student's $t$ test. **e** TIRF images of cells immunostained for KRAP. Typical of 22 cells from 8 independent dishes. **f** Manders split coefficient for IP$_3$R colocalizing with KRAP in different cell regions. Mean ± s.d., $n = 10$ cells for each analysis, $^{**}P < 0.01$, $^{****}P < 0.0001$, one-way ANOVA with Bonferroni's test. **g** z-projection showing colocalization (white) of KRAP and IP$_3$Rs near PM. Scale bar, 5 μm. Typical of 10 cells from 5 independent dishes. **h,** Frequency distribution of centre-to-centre distances between each IP$_3$R punctum (14886 puncta from 22 cells) and nearest KRAP punctum. Observed values and values after randomization of positions of the KRAP puncta (100 iterations) are shown. 28 ± 7% (mean ± s.d., $n = 22$ cells) of IP$_3$R puncta are within 160 nm (red line) of a KRAP punctum (4.2 ± 0.9% after randomization; $P < 0.0001$, Student's $t$ test) (Supplementary Fig. 3e). Scale truncated at 2 μm, beyond which 0.13% of observed values occur. **i,** Confocal section (close to coverslip) of cell stained for actin and KRAP. Typical of 6 cells from 6 independent dishes. **j** Fluorescence intensity profiles (from i) along actin fibres (left) or perpendicular to them (right). Red lines show regions with actin. FU, fluorescence unit. **k** Summary (mean ± s.e.m., 6 cells; ~1500 puncta analysed for each) show distributions of colocalized IP$_3$R-KRAP puncta (centre-centre distances < 160 nm) relative to actin. $^{****}P < 0.0001$, Student's $t$ test. **l** TIRF images show effects of latrunculin A (5 μM, 30 min) on colocalization of IP$_3$R and KRAP with actin filaments. Colocalized IP$_3$R and KRAP puncta associate with actin, even as it depolymerizes to leave residual filaments. Typical of 5 cells from 5 independent dishes.

we showed that IP$_3$Rs and KRAP cluster around the same centroid in each punctum. However, IP$_3$Rs were often quite widely distributed within a punctum, whereas KRAP was more tightly clustered (Fig. 2a–f), suggesting that KRAP tethers a loose cluster of IP$_3$Rs to actin. Loss of KRAP reduced the number of immobile IP$_3$R puncta in the TIRF field, without affecting IP$_3$R expression or the distribution of actin (Fig. 2g–i, Supplementary Fig. 4a–h). Fluorescence recovery after photobleaching (FRAP) analyses confirmed that loss of KRAP reduced the IP$_3$R immobile fraction at the cell periphery, but the immobility of perinuclear IP$_3$Rs was unaffected (Fig. 2j, Supplementary Fig. 4i, j). We conclude that different mechanisms immobilize central and near-PM IP$_3$Rs; only the latter requires KRAP and only these near-PM IP$_3$Rs tethered to actin are licensed to evoke Ca$^{2+}$ puffs[20]. The overall decrease in fluorescence of IP$_3$R puncta in the TIRF field after the loss of KRAP (26 ± 9.7%, Fig. 2k, Supplementary Fig. 4k) and the properties of the remaining puncta, which are more abundant and less bright than in control cells, are consistent with loss of KRAP causing loss of bright puncta tethered beneath the PM, rather than disaggregation of individual IP$_3$R puncta (Fig. 2l, m, Supplementary Fig. 4k–m). We suggest that KRAP, by interacting directly with IP$_3$Rs or with molecules that scaffold the IP$_3$R cluster, tethers one or two pre-assembled IP$_3$R clusters to actin (Fig. 2n).

**KRAP directs IP$_3$Rs to the sites where they are licensed.** We next considered whether KRAP guides IP$_3$Rs to actin immediately beneath the PM or vice versa. There was no difference in the colocalization of near-PM KRAP with actin in cells with and without IP$_3$Rs, suggesting that KRAP can be appropriately targeted without help from IP$_3$Rs (Supplementary Fig. 5a, b). Licensed IP$_3$Rs park immediately beneath the PM and alongside the sites where SOCE occurs, but they are excluded from the narrow MCS where STIM1 and Orai1 interact[20]. The association of immobile IP$_3$Rs with actin through KRAP explains the exclusion of IP$_3$Rs from the MCS because most actin filaments terminate at least 100 nm from SOCE MCS[45]. It does not, however, explain the preferential positioning of licensed IP$_3$Rs alongside SOCE MCS[20]. We used cells with endogenous STIM1 tagged with EGFP (STIM1-EGFP HeLa cells)[46] together with siRNA to reduce expression of all IP$_3$R subtypes to examine any effects of IP$_3$Rs on the subcellular distribution of KRAP and STIM1. Loss of Ca$^{2+}$ from the ER caused STIM1 puncta to accumulate beneath the PM, and these puncta accumulated alongside, but not coincident with, KRAP (Supplementary Fig. 5c–i). The distribution of distances between each STIM1

punctum and the nearest actin-associated KRAP punctum revealed that their preferential affiliation was unaffected by loss of IP$_3$Rs (Fig. 2o). These observations show that it is the association of KRAP with actin that determines the location of immobile IP$_3$Rs near the PM and alongside the ER-PM junctions where SOCE occurs. We conclude that KRAP guides IP$_3$Rs to the sites where they are licensed.

**KRAP is required for Ca$^{2+}$ puffs and global Ca$^{2+}$ signals.** Ca$^{2+}$ puffs provide local Ca$^{2+}$ signals at low stimulus intensities. As stimulus intensities increase and more IP$_3$Rs are primed with IP$_3$, Ca$^{2+}$ signals invade the entire cell as Ca$^{2+}$ waves, the frequency of which increases with stimulus intensity[47]. We confirmed this hierarchy of Ca$^{2+}$ signals in HeLa cells stimulated with histamine to evoke IP$_3$ formation through endogenous pathways (Fig. 3a, Supplementary Fig. 6a). Since all three IP$_3$R subtypes can generate Ca$^{2+}$ puffs[10,11], it has been widely assumed that Ca$^{2+}$ puffs are the building blocks for all IP$_3$-evoked Ca$^{2+}$ signals. However, it is not known whether, during signal propagation, Ca$^{2+}$ puffs recruit only further Ca$^{2+}$ puffs (licensed IP$_3$Rs) or additional IP$_3$Rs (e.g., mobile IP$_3$Rs). Indeed, recent evidence suggests that global Ca$^{2+}$ signals may not be entirely associated with underlying Ca$^{2+}$ puffs[12].

By using TIRFM to record IP$_3$R mobility and Ca$^{2+}$ signals after photolysis of ci-IP$_3$, and then fixing cells to visualize KRAP, we established that Ca$^{2+}$ puffs occur at sites populated by KRAP and immobile IP$_3$R puncta (Fig. 3b–f). Because the centroids of successive Ca$^{2+}$ puffs wander within a site[48–50], reflecting recruitment of sparsely distributed IP$_3$Rs within a punctum (Fig. 2a–d, n, Supplementary Fig. 6b, c), there is some variability in the separation of Ca$^{2+}$ puff centroids and KRAP/IP$_3$R puncta. However, the distances between successive Ca$^{2+}$ puffs at a site, between Ca$^{2+}$ puffs and immobile IP$_3$R, and between Ca$^{2+}$ puffs and KRAP/IP$_3$R puncta are similar (Fig. 3f). This indicates that Ca$^{2+}$ puffs occur at KRAP-associated immobile IP$_3$R puncta (Fig. 3b–f).

In cells with KRAP expression reduced by siRNA, there were very few immobile IP$_3$R puncta, IP$_3$-evoked Ca$^{2+}$ puffs were almost abolished, and the number of sites where Ca$^{2+}$ puffs occurred was massively reduced (Fig. 3g–k, Supplementary Fig. 6d–g, Supplementary Movies 7–9). By adjusting detection thresholds, we confirmed that the lack of Ca$^{2+}$ puffs in KRAP-depleted cells was not a result of failing to detect small Ca$^{2+}$ puffs (Supplementary Fig. 6h, i). We conclude that KRAP is required to both immobilize IP$_3$R clusters on actin and to allow them to evoke Ca$^{2+}$ puffs.

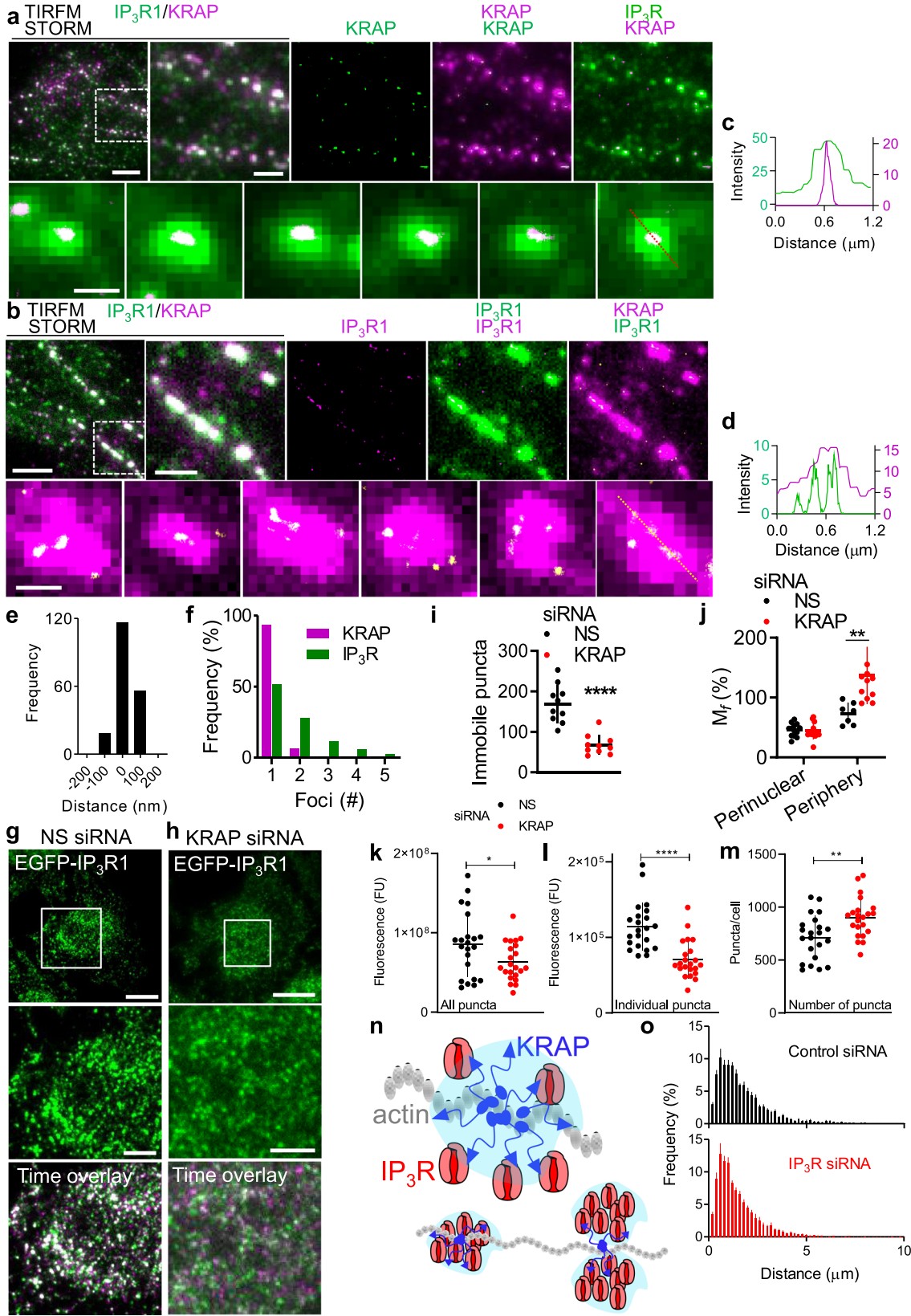

We next considered whether the global Ca²⁺ signals evoked by more intense stimulation also required KRAP. A maximally effective histamine concentration, which evoked large global increases in [Ca²⁺]c in normal cells, caused a negligible increase in [Ca²⁺]c in cells without KRAP (Fig. 3l–n). Nor was there any greater response when the histamine concentration was increased

to one-hundred times that required to evoke a maximal response in normal cells (Fig. 3o). Global Ca²⁺ signals evoked by intense photolysis of ci-IP₃ were also abolished in cells without KRAP (Fig. 3l–n). Neither IP₃R expression nor ER Ca²⁺ content was reduced by KRAP siRNA (Supplementary Fig. 4c, d) indeed the ER Ca²⁺ content was slightly greater in cells without KRAP

**Fig. 2 KRAP tethers immobile IP₃Rs to actin near the PM. a**, **b**, STORM images of KRAP overlying TIRF images of IP₃R (a), and STORM images of IP₃R overlying TIRF images of KRAP (**b**). Scale bars, 5 µm in first image, 2 µm in enlargements of boxed areas, 500 nm in images of individual puncta. Typical of at least 3 experiments. The resolution of our STORM images (FWHM 20–25 nm) is too low to confidently distinguish single IP₃R tetramers (~20 nm across) from tightly packed small clusters. Pixels evident in the enlarged overlays are from the TIRF, rather than STORM, images (pixel sizes 100 nm and 10 nm, respectively). **c**, **d**, Fluorescence intensity profiles across puncta (dashed lines in (**a**) and (**b**)). **e**, **f**, Summary shows distances between centroids of KRAP and IP₃R within each punctum (e, 190 puncta, 3 cells) and distribution of numbers of resolved foci within each punctum (f, 118 IP₃R puncta from 4 cells, 150 KRAP puncta from 3 cells). **g**, **h**, TIRF images of EGFP-IP₃R1 HeLa cells transfected with non-silencing (NS) (**g**) or KRAP siRNA (**h**) and time overlays (30-s interval) showing mobile and immobile IP₃R puncta. Scale bars, 10 µm (5 µm for enlargements). **i**, Summary (mean ± s.e.m., 10 cells from 5 independent dishes, with 41–253 puncta analysed in each) shows numbers of immobile IP₃R puncta per cell. $****P < 0.0001$, Student's $t$ test. **j** Effects of siRNA on mobile fractions ($M_f$) determined by FRAP for peripheral and perinuclear IP₃R puncta (Supplementary Fig. 4i, j). Mean ± s.d., $n = 7$ cells (peripheral, NS siRNA), 12 cells (peripheral and perinuclear, KRAP siRNA), and 13 cells (perinuclear, NS siRNA). $**P < 0.01$ Student's $t$ test. **k–m**, Effects of siRNA on the sum of the fluorescence intensities of all IP₃R puncta in the TIRF field (**k**), the average intensities of individual puncta (**l**), and numbers of puncta (**m**). Mean ± s.d., $n = 22$ cells for each, $****P < 0.0001$, $**P < 0.01$, $*P < 0.05$, Student's $t$ test (**k–m**). **n** KRAP, via its interaction with IP₃Rs or scaffold molecules (pale blue), tethers pre-assembled clusters of sparsely distributed IP₃Rs to actin. **o** Effects of siRNA against the three IP₃R subtypes (10 cells) or NS siRNA (9 cells) on distribution of distances between the centroids of each STIM1 punctum and the nearest actin-associated KRAP punctum in cells treated with thapsigargin ($P < 0.05$, for both distributions relative to distances after randomization of KRAP distribution; Costes randomization test for each cell).

(Supplementary Fig. 4n, o), perhaps a consequence of reduced basal IP₃R activity. The effects of siRNA-mediated knockdown of KRAP expression on IP₃R mobility and Ca²⁺ puffs were reversed by expression of a siRNA-resistant KRAP (Fig. 3g–k). We conclude that all cytosolic Ca²⁺ signals, local and global, require IP₃Rs licensed by their association with KRAP.

**Endogenous KRAP determines the sensitivity of cells to IP₃.** Only a small fraction of IP₃Rs is licensed to respond. We therefore considered whether KRAP might determine the number of licensed IP₃Rs. Over-expressing KRAP in EGFP-IP₃R1 HeLa cells (to 9.5-times endogenous levels, Supplementary Fig. 6j) increased the number of immobile IP₃R puncta without increasing the number of IP₃Rs within a punctum (Fig. 4a–c). We conclude that even though not all KRAP is associated with IP₃Rs (Fig. 1e), the availability of endogenous KRAP controls the number, rather than the size, of immobile IP₃R puncta. This aligns with evidence that KRAP captures pre-existing IP₃R clusters and tethers them to actin (Fig. 2k–m, Supplementary Fig. 4k–m). Consistent with that interpretation, over-expressed KRAP increased the frequency of Ca²⁺ puffs evoked by photolysis of ci-IP₃ and the number of sites where they occurred, but not the amplitude of individual Ca²⁺ puffs (Fig. 4d–f, Supplementary Movie 10). We confirmed that our identification of more Ca²⁺-release sites was not due to improved detection efficiency arising from more frequent Ca²⁺ puffs (Supplementary Fig. 6k). The increased activity caused Ca²⁺ signals to propagate globally after much shorter intervals and to evoke larger increases in [Ca²⁺]ꞓ (Fig. 4g). Even in cells stimulated with a maximal histamine concentration, KRAP over-expression caused the peak Ca²⁺ signal to increase by 23 ± 8%. We conclude that endogenous KRAP limits the number of licensed IP₃Rs and thereby the ability of a cell to evoke cytosolic Ca²⁺ signals.

**Discussion**
Ca²⁺ puffs are elementary regenerative events evoked by IP₃ in mammalian cells; yet while most IP₃Rs are mobile, Ca²⁺ puffs occur repeatedly at the same sites within a cell[16–18]. We have now resolved this long-standing conundrum by identifying an additional level of IP₃R regulation that precedes IP₃R gating by IP₃ and Ca²⁺. We have shown that KRAP licenses IP₃Rs to respond by tethering loose confederations of IP₃Rs to actin filaments immediately beneath the PM and alongside the sites where SOCE occurs (Fig. 5a, b). All IP₃-evoked cytosolic Ca²⁺ signals, whether local (Ca²⁺ puffs) or global, require licensing of IP₃Rs by KRAP

(Fig. 5a, b). Licensing by KRAP is probably a feature of all IP₃Rs since all three IP₃R subtypes interact with KRAP[29,35,36], and loss of KRAP abolishes IP₃-evoked Ca²⁺ signals in HeLa cells (Fig. 3l–o), which express all three IP₃R subtypes[20].

How can an obligatory need for KRAP in intact cells be reconciled with evidence that IP₃Rs can open under experimental conditions where their association with KRAP or actin is unlikely? Such interactions are impossible, for example, after functional reconstitution of purified IP₃R protein[51] or in patch-clamp recordings[52], and they are unlikely in permeabilized cells;[3] yet in each case, IP₃ evokes opening of IP₃Rs. We suggest that IP₃Rs in intact cells are probably constitutively repressed, and that licensing by KRAP relieves the inhibition. We have not yet explored the molecular basis of the repression. IRBIT (IP₃R-binding protein released by IP₃)[53], Bcl2 (B-cell lymphoma 2)[54], annexin A1[55] and many other signals inhibit IP₃Rs[56], and they may be lost when cells are disrupted, perhaps thereby alleviating the need for KRAP in analyses of broken cells.

A second issue arises from observations that not all IP₃-evoked Ca²⁺ signals in all cells arise immediately beneath the PM. We consider two possible explanations. It may be that in other cells or in response to appropriate regulatory signals, IP₃Rs may be licensed by association with KRAP in different subcellular locations, allowing them to initiate Ca²⁺ signals away from the PM. Alternatively, and consistent with recent analyses of local and global Ca²⁺ signals in HEK cells[12], a flurry of Ca²⁺ puffs mediated by licensed IP₃Rs may provide a signal (perhaps mediated by a decrease in ER luminal [Ca²⁺])[12] that ignites the activity of unlicensed IP₃Rs. Hence, and consistent with our observations (Fig. 3), all IP₃-evoked Ca²⁺ signals would require licensing of IP₃Rs by KRAP: either directly to evoke Ca²⁺ puffs, or indirectly (through the prior occurrence of Ca²⁺ puffs) to evoke global Ca²⁺ signals. The second scheme also aligns with evidence from pancreatic acinar cells. In these cells, KRAP is located at the apical pole of each cell[57], where IP₃-evoked Ca²⁺ signals initiate before propagating via IP₃Rs and ryanodine receptors to the basal pole;[58] disruption of the actin cytoskeleton perturbs these local Ca²⁺ signals and the apical distribution of IP₃Rs[26].

Licensing of IP₃Rs by KRAP is likely to have further implications. By positioning responsive IP₃R clusters alongside the MCS where SOCE occurs, KRAP may ensure that activation of IP₃Rs selectively depletes the ER that is best placed to regulate SOCE. We speculate that this may allow substantial local depletion of small ER compartments close to the PM and thereby allow local digital regulation of SOCE (Fig. 5d)[20,59]. In this way, the substantial ER depletion required to effectively activate STIM1[60] may

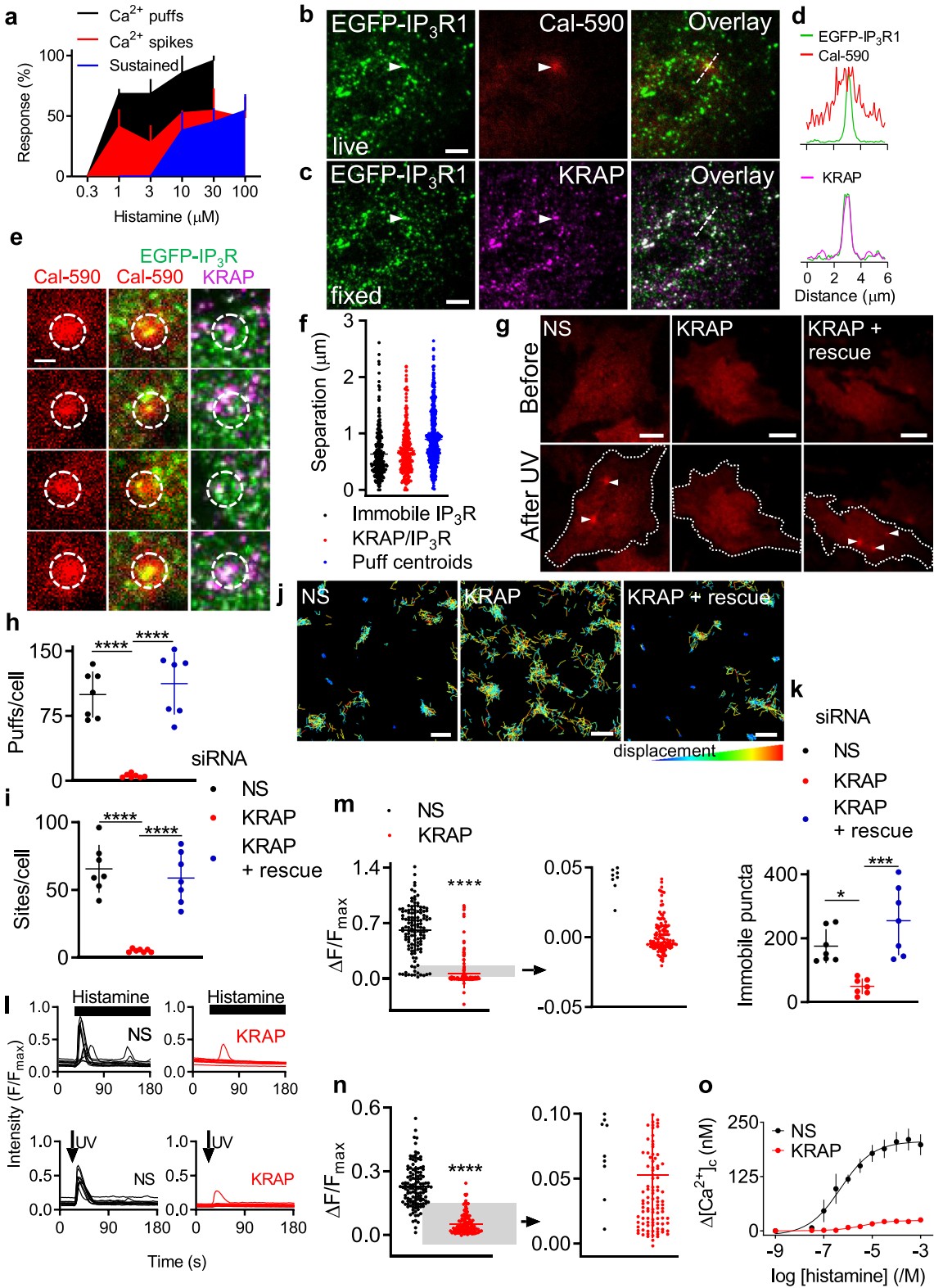

be achieved without massive loss of $Ca^{2+}$ from the entire ER, which might otherwise compromise other ER functions like protein folding. KRAP takes its name from the discovery that it is up-regulated in many KRas-transformed cells[33]. Licensing of IP$_3$Rs by KRAP may, therefore, provide another site of interaction between endogenous Ras and $Ca^{2+}$ signalling[61]. Finally, the

membrane lipid phosphatidylinositol 4,5-bisphosphate (PIP$_2$), is both the source of IP$_3$ and a key regulator of actin filaments[62], suggesting that PIP$_2$ may control both licensing of IP$_3$Rs and their activation by IP$_3$ (Fig. 5c).

We conclude that licensing of IP$_3$Rs by KRAP allows cells to respond to IP$_3$ and determines the spatial organization of

**Fig. 3 KRAP is required for IP$_3$-evoked Ca$^{2+}$ signals. a** Hierarchical recruitment of Ca$^{2+}$ signals by histamine (Supplementary Fig. 6a). **b** TIRF images of part of an EGFP-IP$_3$R1 HeLa cell showing IP$_3$Rs and a Ca$^{2+}$ puff (arrow) evoked by photolysis of ci-IP$_3$, and their overlay. **c** Same cell after KRAP immunostaining shows IP$_3$Rs, KRAP and overlay. Arrow highlights location of Ca$^{2+}$ puff. Scale bars (**b**, **c**), 5 μm. Images typical of 3 cells from 3 independent experiments (**b**, **c**). **d** Fluorescence intensity profiles along dashed lines in (**b**) and (**c**). **e** Examples of colocalized KRAP/IP$_3$R puncta and Ca$^{2+}$ puffs evoked by photolysis of ci-IP$_3$. Scale bar, 2 μm. Typical of 3 cells from 3 independent experiments. **f** Centre-centre distances measured from Ca$^{2+}$ puffs to immobile IP$_3$Rs (black, $n = 3$ cells), IP$_3$R/KRAP puncta (red, $n = 3$ cells), or between centres of successive Ca$^{2+}$ puffs at the same site (blue, $n = 19$ cells) (Supplementary Fig. 6b, c). Individual values with mean ± s.d for the indicated number of cells. **g** TIRF images of Ca$^{2+}$ puffs evoked by photolysis (150 ms) of ci-IP$_3$ after the indicated treatments. Scale bars, 10 μm. **h**, **i**, Summary (mean ± s.d, 7 cells, with 4–152 puffs detected per cell). $^{****}P < 0.0001$, one-way ANOVA with Bonferroni's test. **j** Single-particle trajectories (coloured according to frame-to-frame displacement) of EGFP-IP$_3$R1 puncta in cells treated with NS or KRAP siRNA alone or with expression of siRNA-resistant KRAP (rescue). Scale bars, 500 nm. **k** Summary (mean ± s.d., 7 cells with 16–407 puncta per cell analysed) shows numbers of immobile IP$_3$R puncta per cell. $^{*}P < 0.05$, $^{***}P < 0.001$, one-way ANOVA with Bonferroni's test. **l** Randomly selected traces (10 cells for each) show responses from EGFP-IP$_3$R1 HeLa cells treated with NS or KRAP siRNA and stimulated with a maximally effective concentration of histamine (100 μM) or intense photolysis of ci-IP$_3$ (600 ms). **m**, **n** Summary (mean ± s.d.) shows increases in [Ca$^{2+}$]$_c$ (reported as ΔF/F$_{max}$) evoked by histamine (m, $n = 157$ and 125 cells from 7 and 6 independent siRNA treatments for KRAP and NS siRNA, respectively) or photolysis of ci-IP$_3$ (n, $n = 109$ and 127 cells from 6 and 7 independent siRNA treatments for KRAP and NS siRNA). Enlargements around ΔF/F$_{max} = 0$ highlight many unresponsive KRAP siRNA-treated cells. $^{****}P < 0.0001$, Student's $t$ test. Different Ca$^{2+}$ indicators were used with histamine (Cal-590, $K_d = 561$ nM) and ci-IP$_3$ (Calbryte 590, $K_d = 1400$ nM). **o** Effect of NS and KRAP siRNA on peak increases in [Ca$^{2+}$]$_c$ (Δ[Ca$^{2+}$]$_c$) evoked by the indicated concentrations of histamine in populations of EGFP-IP$_3$R1 HeLa cells stimulated in HBS. Mean ± s.e.m., $n = 4$, each with 3 replicates.

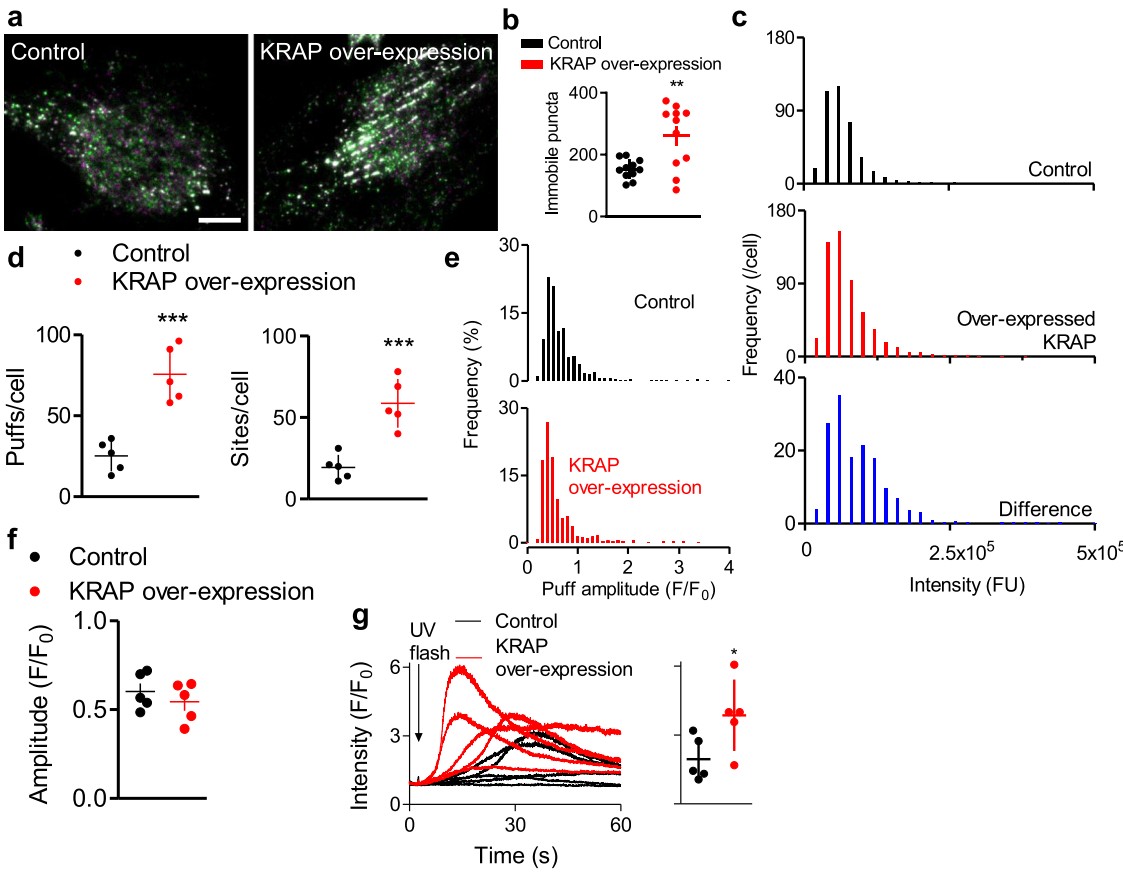

**Fig. 4 Endogenous KRAP determines the number of licensed IP$_3$Rs. a** TIRFM images of EGFP-IP$_3$R1 at 30-s intervals (green and magenta) show more immobile IP$_3$R puncta (white) in cells over-expressing KRAP. Scale bar, 10 μm. **b** Summary (mean ± s.d., 11 cells) show numbers of immobile IP$_3$R puncta per cell. $^{**}P < 0.01$, Student's $t$ test. **c** Fluorescence intensity distributions for EGFP-IP$_3$R in control and cells over-expressing KRAP (4762–6559 puncta from 12 cells). The difference plot confirms that KRAP over-expression does not affect the size of IP$_3$R puncta. **d** Numbers of Ca$^{2+}$ puffs and sites detected during the 5.5-s interval after photolysis of ci-IP$_3$ (individual values, mean ± s.d., 5 cells). $^{***}P < 0.001$, Student's $t$ test. **e**, Distribution of Ca$^{2+}$ puff amplitudes (61–142 Ca$^{2+}$ puffs in each of 5 cells). **f** Summary results (mean ± s.e.m., 5 cells) show mean puff amplitudes. $P > 0.05$, Student's $t$ test. **g** Examples of whole-cell Cal-590 fluorescence changes evoked by photolysis (150 ms) of ci-IP$_3$ in control and KRAP-over-expressing cells. Histogram shows peak amplitudes (mean ± s.d. $n = 5$ cells). $^{*}P < 0.05$, Student's $t$ test.

cytosolic Ca$^{2+}$ signals. Licensing may allow dynamic actin filaments to regulate Ca$^{2+}$ signalling, and IP$_3$Rs to regulate SOCE through local depletion of ER Ca$^{2+}$ stores (Fig. 5)[20,59].

## Methods

**Materials.** Cal-590 AM, Calbryte 590 AM and Fluo-8 AM were from AAT Bioquest (Sunnyvale, CA, USA). A membrane-permeant form of caged-IP$_3$ (ci-IP$_3$/PM: D-2,3-O-isopropylidene-6-O-(2-nitro-4,5-dimethoxy)benzyl-*myo*-inositol 1,4,5-trisphosphate hexakis(propionoxymethyl) ester) was from SiChem (Bremen,

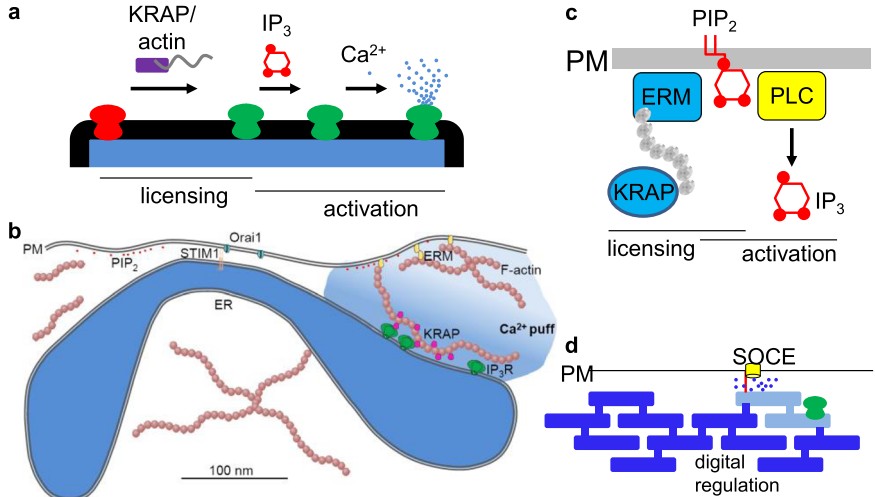

**Fig. 5 KRAP licenses IP$_3$Rs to evoke Ca$^{2+}$ signals. a** IP$_3$-evoked Ca$^{2+}$ release requires licensing by the association of IP$_3$Rs with KRAP and actin. IP$_3$R activation is then triggered by sequential binding of IP$_3$ and Ca$^{2+}$. **b** Organization of IP$_3$Rs near SOCE junctions, drawn to approximate scale. **c** PIP$_2$ may regulate both IP$_3$R licensing by controlling assembly of actin filaments (through ezrin, radixin and moesin (ERM), for example) and as the substrate from which IP$_3$ is produced by PLC. **d**, Local release of Ca$^{2+}$ from ER by licensed IP$_3$Rs immediately beneath the PM may allow digital regulation of SOCE.

Germany). Acti-stain 670 phalloidin (# PHDN1-A) was from Cytoskeleton (Denver, CO, USA). BAPTA (1,2-bis(o-aminophenoxy)ethane-N,N,N′,N′-tetraacetic acid) was from Molekula (Darlington, UK). Histamine was from Sigma-Aldrich. Bovine serum albumin (BSA) was from Europa Bio-Products (Ely, UK). Withaferin A was from AdipoGen Life Sciences (Liestal, Switzerland). AlexaFluor-568 phalloidin and restriction enzymes were from ThermoFisher. Human fibronectin was from Merck Millipore (Watford, UK). Plasmids encoding the following proteins were from: LifeAct-mCherry-N1 (Addgene #40908)[63], LifAct-7-iRFP670 (Addgene #103032)[64], human KRAP with an N-terminal Myc-DDK tag (OriGene, Rockville, MD, USA, #RC205550), mCherry-keratin (Addgene #55066) and mCherry-vimentin-N-18 (Addgene #55158, deposited by Michael Davidson, Florida State University, FL, USA). Cytochalasin D and latrunculin-A were from Tocris (Abingdon, UK). Antibodies (for Western blotting, WB; or immunocytochemistry, IC) were from: β-actin (mouse monoclonal; WB 1:10000; Cell Signaling Technology, Leiden, Netherlands, #8H10D10, undefined clone #); KRAP (rabbit polyclonal; WB 1:1000; IC 1:400; ProteinTech, Manchester, UK, #14157-1-AP), the same KRAP primary antibody was custom-conjugated to YF-594 for TIRF imaging during STORM analyses of IP$_3$Rs (1:400, ProteinTech); GFP Tag-AlexaFluor-647 (STORM 1:400, ThermoFisher, #31852); IP$_3$R1 (rabbit, raised against a C-terminal peptide 2732–2750 of rat IP$_3$R1; WB 1:1000, Merck Millipore, #AB5882); IP$_3$R2 (rabbit, custom-made to a C-terminal peptide GFLGSNTPHEN HHMPPH; WB 1:500, IC 1:200, Pocono Rabbit Farm and Laboratory);[65] IP$_3$R3 (mouse monoclonal; WB 1:1000, IC, 1:200, BD Transduction Laboratories, Wokingham, UK, #610313, clone 2); STIM1 (rabbit monoclonal; WB 1:800, Cell Signaling Technology, #5668, undefined clone #); vimentin (chicken polyclonal; IC 1:1000, Novus Biologicals, Centennial, CO, USA, #NB300-223); GFP-Booster Atto488 (IC 1:500, Chromotek, Planegg-Martinsried, Germany #gba488); donkey anti-rabbit IgG-HRP (WB 1:5000, SantaCruz, Heidelberg, Germany, SC-2313); donkey anti-mouse IgG-HRP (WB 1:2000, Santa Cruz, SC-2314); goat anti-rat IgG-HRP (WB 1:5000, Santa Cruz, SC-2020); goat anti-rabbit AlexaFluor-594 (IC 1:1000, ThermoFisher, #A11012); goat anti-rabbit AlexaFluor-647 (IC and STORM 1:500, ThermoFisher, #A21244); goat anti-mouse AlexaFluor-568 (IC 1:1000, ThermoFisher, #A11004); and goat anti-chicken AlexaFluor-568 (IC 1:1000, ThermoFisher, #A11041). Additional sources of materials are provided in relevant methods.

**Cell culture and transfection**. HEK cells, HEK cells devoid of all IP$_3$Rs (HEK-3KO, Kerafast, Boston, MA, USA)[8], HeLa cells, STIM1-EGFP HeLa cells[66] and EGFP-IP$_3$R1 HeLa cells[20] were cultured in Dulbecco's modified Eagle's medium/F-12 with Gluta-MAX (ThermoFisher, Waltham, MA, USA) supplemented with foetal bovine serum (FBS, 10%, Sigma-Aldrich, Gillingham, Dorset, UK). The cells were maintained at 37 °C in humidified air with 5% CO$_2$ and passaged every 3-4 days using Gibco TrypLE Express (ThermoFisher). We have reported an extensive characterization of the EGFP-IP$_3$R1 HeLa cell line, in which monomeric EGFP was attached to the N-terminal of all copies of the endogenous IP$_3$R1 gene (*ITPR1*) using TALENs (transcription activator-like effector nucleases)[20]. The characterization included evidence that EGFP-IP$_3$R1 is functional. In the STIM1-EGFP HeLa cell line, CRISPR/Cas9 was used to add monomeric EGFP to the C-terminal of one of the two *STIM1* genes[66]. For STIM1-EGFP HeLa cells, we have confirmed that the only fluorescent protein in the cells is STIM1-EGFP and that histamine-evoked Ca$^{2+}$ signals and SOCE are unperturbed by the gene-editing[46]. KRAP is up-regulated in some cells with mutant

KRas[33], but we confirmed by pyrosequencing (Source Bioscience, Nottingham, UK) that there were no mutations in codons 12, 13 (exon 2), 59, 61 (exon 3), 117 or 146 (exon 4) of the *KRAS* gene in the EGFP-IP$_3$R1 HeLa cells.

For imaging, cells were grown on 35-mm glass-bottomed dishes (Cellvis, IBL Baustoff+Labor GmbH, Gerasdorf bei Wein, Austria) coated with human fibronectin (10 μg ml$^{-1}$). Regular screening confirmed that all cells were free of mycoplasma. The authenticity of the EGFP-IP$_3$R1 HeLa cells (Eurofins, London, UK) and HEK-3KO cells (DNA Diagnostic Center, Fisher Scientific) was verified by short-tandem repeat profiling.

For transient transfections, EGFP-IP$_3$R1 HeLa cells grown on Cellvis imaging dishes were transfected with the appropriate plasmid (1–2 μg DNA per dish) using ViaFect (Promega, Madison, WI, USA) transfection reagent (3 μl per 1 μg DNA) according to the manufacturer's instructions. Cells were used after 24 h.

**Western blotting**. Cells grown on 75-cm$^2$ culture flasks were harvested using either enzyme-free cell dissociation buffer (ThermoFisher) or, for siRNA experiments, by scraping cells into lysis medium (LM, 150 mM NaCl, 0.5 mM EDTA, 1% Triton X-100, 10 mM Tris/HCl, pH 7.5) containing a protease inhibitor mini-tablet with EDTA (Pierce, 1 tablet per 10 ml). Cells were then lysed by incubation in LM at 4 °C for 1 h, sonicated (Transonic ultrasonic bath, 3 × 10 s), and the supernatant was collected (20,000 × *g*, 30 min). Since actin-associated proteins are more resistant to detergent-extraction, and KRAP facilitates IP$_3$R association with actin (Fig. 1), the Triton X-100 in LM was supplemented with 0.5% sodium deoxycholate and 0.1% sodium dodecyl sulphate (SDS)[29] for western blot (WB) analyses of IP$_3$Rs before and after KRAP knock-down (Supplementary Fig. 4c, d).

For WB, proteins were separated using NuPAGE 3–8% Tris-acetate gels (ThermoFisher) and transferred onto an iBlot PVDF membrane using an iBlot gel-transfer device (ThermoFisher). The PVDF membrane was blocked by incubation (1 h, 20 °C) in TBST, which comprised Tris-buffered saline (TBS: 137 mM NaCl, 20 mM Tris, pH 7.6) supplemented with bovine serum albumin (BSA, 5%) and Tween-20 (0.1%). The blocked membrane was incubated (16 h, 4 °C) with primary antibody (in TBST with 1% BSA), washed (3 × 5 min) in TBST, and incubated (1 h, 20 °C) with horseradish peroxidase (HRP)-conjugated secondary antibody in TBST with 1% BSA. The antibodies used and their dilutions are listed in the Materials section. After further washes with TBST (3 × 5 min), HRP was detected using ECL Prime Western blotting detection reagents (GE Healthcare Life Sciences, Little Chalfont. UK) and a PXi chemiluminescence detection system (Syngene, Cambridge, UK). Where main figures show only part of a WB, the complete WB is shown in Supplementary Fig. 7.

**Treatment with siRNA**. Cells grown in either glass-bottomed imaging dishes or multi-well plates (6 or 96 wells) were transfected with siRNA directed against KRAP (50 nM, #AM16708 or #4392420, ThermoFisher) or a non-silencing (NS) control siRNA (50 nM, #AM4611, ThermoFisher) using either siPORT NeoFX transfection reagent (ThermoFisher, 220 ng siRNA per μl reagent) or a Neon transfection system (ThermoFisher) according to the manufacturer's instructions. The two KRAP siRNAs were used interchangeably and with indistinguishable knockdown efficiencies (Supplementary Fig. 4a, b). Cells were used after 72 h. We confirmed by immunostaining that after treatment with KRAP siRNA, KRAP was undetectable in 90 ± 0.1% of cells (55 cells from 3 independent experiments) and 0 ± 0 % for cells treated with NS siRNA (53 cells from 3 experiments). The very small

residual histamine-evoked $Ca^{2+}$ signals (~11%, Fig. 3o) in cell populations treated with KRAP siRNA are likely attributable to the 10% of cells with residual KRAP. Images of cells treated with either siRNA were randomly selected for single-cell analyses.

For siRNA-mediated knockdown of the three $IP_3R$ subtypes, cells were simultaneously transfected with siRNA to each $IP_3R$ subtype (40 nM of each FlexiTube siRNA, Qiagen, Hilden, Germany: $IP_3R1$, #S100034545; $IP_3R2$, #S1000 34552; $IP_3R3$, #S100034580) or NS siRNA (120 nM, Qiagen #1027281). Cells were used after 72 h.

A siRNA-resistant version of the KRAP plasmid was prepared by digesting the plasmid encoding human KRAP (Origene, #RC205550) at Bst1107I and ScaI restriction sites. This was followed by the introduction of a DNA string of the excised region containing silent mutations at the siRNA target site with 30 bp flanking regions of the two restriction sites (ThermoFisher), into the plasmid using Gibson assembly (Gibson Assembly Master Mix, New England Biolabs, Ipswich, MA, USA). The sequence of the DNA from which the siRNA-targeted mRNA sequence derives (bp 2011–2029 of the coding sequence) was: GCTAAATGCAGT GATATGA in the native DNA, and GCGAAGTGTTCAGACATGA in the mutated form (mutated residues are underlined). The modified and native DNAs encode the same protein sequence.

**Immunocytochemistry.** Cells grown on fibronectin-coated Cellvis glass-bottomed dishes were washed three times in phosphate-buffered saline (PBS, 20 °C), fixed with paraformaldehyde (4%) in PBS (30 min), washed three-times in PBS, and permeabilized by incubation (5 min) in PBS containing Triton X-100 (0.25%). After three washes with PBS, cells were incubated (1 h) in PBS containing BSA (5%), washed briefly, and then incubated (1 h) with primary antibody (details in Materials section) in PBS containing BSA (3%). After three washes with PBS, cells were incubated (1 h) with an appropriate AlexaFluor-conjugated secondary antibody, fluorescent phalloidin or GFP-Booster (details in Materials section and legends) in PBS containing BSA (3%). The cells were then washed three times with PBS before imaging. For matched analyses of live and immunostained cells (Fig. 3b–f), fixation, permeabilization and immunostaining were performed without moving the sample from the microscope stage.

**Fluorescence microscopy.** Fluorescence microscopy used an inverted Olympus IX83 microscope equipped with a 100× oil-immersion TIRF objective (numerical aperture, NA 1.49), a multi-line laser bank (488 nm, 561 nm, 638 nm and 647 nm) and an iLas2 targeted laser illumination system (Cairn, Faversham, UK). In some experiments that required larger fields of view, we used a ×60 (NA 1.45) oil-immersion TIRF objective (Fig. 3l–n). Excitation light was transmitted through a quad dichroic beam splitter (TRF89902-QUAD, Chroma). Emitted light was passed through appropriate filters (Cairn Optospin; peak/bandwidth: 525/50, 630/75 and 700/75 nm) and detected with either an iXon Ultra 897 electron multiplying charge-coupled device (EMCCD) camera (512 × 512 pixels, Andor) or a Prime 95B scientific complementary metal oxide semiconductor (sCMOS) camera (1200 × 1200 pixels, Photometrics). Spinning-disc confocal microscopy (SDCM) used a spinning disc with a 70-μm pinhole (X-Light, CrestOptics, Rome, Italy). For total internal reflection fluorescence microscopy (TIRFM), the penetration depth was 90–120 nm. The iLas2 illumination system was used for TIRFM and FRAP.

Super-resolution confocal microscopy used an inverted Nikon Eclipse Ti2 microscope equipped with a ×100 oil-immersion TIRF objective (NA 1.49), an X-Light V3 spinning-disc confocal unit with 50-μm pinholes (CrestOptics), and a Live-SR super-resolution module (Gataca Systems, Massy, France). Excitation light (470, 555 and 640 nm) from a laser diode illuminator (89 North, Williston, VT, USA) was transmitted through a quad-band filter set (405/470/555/640 nm). Emitted light was passed through an appropriate emission filter (GFP, TRITC or Cy5) and detected with a Prime 95B camera. Super-resolution was achieved by multifocal structured illumination[67] using the Live-SR module and optical reassignment processing to give a lateral resolution of approximately 120 nm.

Before analysis, all fluorescence images were corrected for background by subtraction of fluorescence detected from a region outside the cell. Image capture and processing used MetaMorph Microscopy Automation and Image Analysis Software (version 7.10.1.161, Molecular Devices, San Jose, CA, USA) and Fiji (https://fiji.sc)[68], respectively. Confocal images were deconvolved using the Microvolution deconvolution algorithm (version 2015.05)[69]. All images are presented in RGB colour (16-bit) format.

**Stochastic optical reconstruction microscopy (STORM).** The methods used to generate STORM images were exactly as described[20]. Cells were fixed in PBS containing paraformaldehyde (4%, 30 s), permeabilized and immunostained. For identification of KRAP by STORM, we used a rabbit polyclonal primary antibody (1:400) and goat anti-rabbit AlexaFluor-647 secondary antibody (1:500); $IP_3Rs$ were identified in TIRF from their EGFP fluorescence. To identify $IP_3Rs$ using STORM, EGFP-$IP_3R1$ was visualized using GFP Tag-AlexaFluor-647 (1:400); KRAP was identified in TIRF using the same primary antibody used for STORM analyses, but conjugated directly to YF-594 (custom-made by ProteinTech, Manchester, UK). The medium used for STORM comprised: Tris-HCl (50 mM), pH 8.0, NaCl (10 mM), glucose (555 mM), catalase (Sigma-Aldrich, 34 μg ml$^{-1}$),

glucose oxidase (Sigma-Aldrich, 560 μg ml$^{-1}$), 2-mercaptoethanol (Sigma-Aldrich, 1%) and cyclooctatetraene (Sigma-Aldrich, 1%). For STORM imaging, stochastic blinking of individual fluorophores was achieved by first bleaching cells in semi-TIRF mode using full laser power (647 nm). Images (256 × 256 pixels, 20,000 frames, 50 frames per s) were then acquired in TIRF at a lower laser power (647 nm) using a TIRF objective (×100, NA = 1.49, with an intermediate magnification of ×1.6) and an Andor iXon Ultra EMCCD camera. Image acquisition and fitting used WaveTracer (MetaMorph) to detect genuine blinking events followed by fitting a centroid to each blinking event within each 10 nm × 10 nm pixel. Point localizations of blinking events from each of the 20,000 frames were collated to construct a super-resolution STORM image[20]. STORM images generated using four-colour TetraSpeck microspheres (diameter 0.1 μm, ThermoFisher) confirmed that there was no chromatic aberration or drift during image acquisition. The lateral resolution for STORM (full-width at half-maximal amplitude, FWHM) determined using TetraSpeck microspheres was of 21.8 ± 4.0 nm, and 24.2 ± 5.6 nm when determined using STORM images generated with AlexaFluor 647.

**Fluorescence recovery after photobleaching (FRAP).** The methods used for wide-field FRAP measurements, after photobleaching a circular region of interest (ROI), were exactly as described previously[20]. Briefly, time-lapse images (~1 frame per second) were acquired in epifluorescence mode using a 488-nm laser. A circular ROI (diameter = 2.3 μm) was rapidly photobleached by raster scanning at either a perinuclear or peripheral region of a cell, using a 395-nm laser and an iLas2 laser illumination system. Recovery of fluorescence was then recorded. Images were background-corrected by subtracting the fluorescence recorded from a region outside the cell. The mobile fraction was calculated using the ImageJ plugin, FRAP profiler (http://worms.zoology.wisc.edu/ImageJ/FRAP_Profiler.java).

**Colocalization analyses.** These analyses required different methods to accommodate our need to quantify relationships between fluorophores where only subsets of particles were colocalized (e.g., KRAP and $IP_3R$); the need to compare two or three colocalized fluorophores; our use of TIRF and spinning-disc confocal microscopy, where the latter contributes more out-of-focus fluorescence; and our need to compare both punctate (e.g., KRAP and $IP_3R$) and elongated (e.g., actin) structures, where only the punctate structures are amenable to object-based analyses.

Since only subpopulations of $IP_3R$ and KRAP puncta colocalize, we used an object-based method (DiAna, version 1.1)[44] to quantify their colocalization by measuring centre-to-centre distances between $IP_3R$ and KRAP puncta (Fig. 1h, k, Supplementary Fig. 2a–d, 3e). The same methods were used to determine separations of KRAP and STIM1 puncta (Fig. 2o). The separations between the centroids of $Ca^{2+}$ puffs and either immobile $IP_3R$, KRAP/$IP_3R$ or the centroids of successive $Ca^{2+}$ puffs were manually determined (Fig. 3f). Background-corrected TIRF images of fluorescence from EGFP-$IP_3R$ and immunostained KRAP were segmented using a spot segmentation method, which identifies all local intensity maxima in a frame and then uses a threshold to select the local maximum for each object. The algorithm then computes the radial distribution of pixel intensities around each local maximum to define a border intensity threshold for the objects. It then progresses radially from each local maximum and includes pixels if their intensities are both above the border intensity threshold and below the intensity of the pixel that was last accepted; acceptance also requires that a pixel is adjacent to other accepted pixels. Centre-to-centre distances between the segmented spots ($IP_3R$ and KRAP puncta, which need not be circular) are then calculated. Since these measurements provide sub-pixel localization accuracy, we considered $IP_3R$ and KRAP puncta to be colocalized if their centre-to-centre distance was <160 nm (the width of a single pixel in all analyses, except for super-resolution confocal microscopy where the pixel size was 65 nm and criterion for colocalization was a separation of <130 nm). Our aim was to use similar criteria for colocalization for each analysis, while ensuring that the number of pixels used to define colocalization for each imaging method came closest to the theoretical lateral resolution of the method. To assess the statistical significance of the colocalization, the algorithm randomizes (100 iterations) the segmented image of all KRAP puncta before re-calculating the centre-to-centre distances[44]. This analysis was used to provide the cumulative distribution of centre-to-centre distances for unperturbed $IP_3R$ puncta and randomly distributed KRAP puncta with its 95% confidence interval (CI) (Supplementary Fig. 3e). The same approach was applied to all object-based colocalization analyses to establish their statistical significance.

Colocalization analyses of $Ca^{2+}$ puffs with EGFP-$IP_3Rs$ again compared distances between centroids, but using coordinates for $Ca^{2+}$ puffs derived from FLIKA (see the section on Analysis of $Ca^{2+}$ puffs) and coordinates for EGFP-$IP_3R$ puncta derived from Fiji.

For colocalization analyses of actin (which are not amenable to object-based methods) with $IP_3R$ or KRAP, actin images were first Gaussian-filtered ($\sigma = 0.5$) to remove uneven background fluorescence. We then used the Fiji JACoP plugin[70] to calculate the Mander's split coefficient, which reports the fraction of $IP_3Rs$ or KRAP colocalized with actin (Fig. 1d, Supplementary Fig. 5b).

For confocal sections away from the PM, object-based analyses were impracticable because the increased out-of-focus fluorescence prevented reliable identification of local fluorescence maxima with DiAna. To quantify colocalization of $IP_3R$ and KRAP puncta in confocal sections, deconvolved images were Gaussian-

filtered ($\sigma = 0.5$) to remove noise. For confocal sections remote from the coverslip, a peripheral region of interest (ROI) (i.e. close to the PM) was defined by an annulus extending 15 pixels (i.e. 2.4 µm) inward from the cell boundary, while the remaining core formed the central ROI (Fig. 1f). For the confocal section closest to the coverslip (i.e. near-PM; similar to the TIRF field), the ROI was defined by the entire region enclosed by the cell boundary. Colocalization between IP$_3$R and KRAP puncta was measured (Mander's split coefficient for IP$_3$R) using the JACoP plugin[70] for each ROI (Fig. 1f).

Since methods for quantifying colocalization of more than two fluorophores by object-based colocalization methods are not well developed, we measured the colocalization of IP$_3$R, KRAP and actin using a mask to define regions of the cell containing actin filaments, and compared them with regions without actin (Fig. 1k). For each cell, an area of 576–768 µm$^2$ (~30% of the cell area) was used for analysis. Similar methods were used for analyses of vimentin filaments. Deconvolved confocal images were Gaussian-filtered for actin staining ($\sigma = 0.5$). Centre-to-centre distances between IP$_3$R and KRAP puncta were then measured using the DiAna plugin[44] for ROI within and outside the actin mask. We considered IP$_3$R and KRAP puncta to be colocalized if their centre-to-centre distance was <160 nm (< 130 nm for super-resolution confocal microscopy).

**Analysis of Ca$^{2+}$ signals in single cells.** For analysis of Ca$^{2+}$ puffs, EGFP-IP$_3$R1 HeLa cells were grown on fibronectin-coated Cellvis dishes and transfected with siRNA (see the section on Treatment with siRNA) or plasmid encoding KRAP (see the section on Cell culture and transfection). Before use, cells were washed twice with HBS and incubated (1 h) in HBS containing Cal-590-AM (2 µM, 37 °C) or Calbryte 590-AM (2 µM, 20 °C) and ci-IP$_3$/PM (1 µM), washed twice with HBS, and then incubated (30 min, 20 °C) in HBS containing EGTA-AM (5 µM). EGTA is a slow Ca$^{2+}$-buffer that limits regenerative propagation of global Ca$^{2+}$ signals without perturbing Ca$^{2+}$ puffs[71]. After two further washes, cells were incubated in HBS (30 min 20 °C), washed and imaged in HBS at 20 °C. Fluorescence was recorded for 32.5 s by TIRFM (561-nm excitation, 630/75 nm emission, 20 frames per s) with illumination for 25 ms in every 50-ms capture interval to minimize photobleaching. A flash of ultraviolet (UV) light (150–600 ms, 395/20 nm, SPECTRA X-light engine, Lumencor, Beaverton, OR, USA) was delivered after 2.5 s to photolyse ci-IP$_3$. Images were collected using MetaMorph, corrected for background fluorescence, and Ca$^{2+}$ puffs were detected and analysed using the FLIKA algorithm (version 1)[72]. Similar methods (using Cal-590, but without ci-IP$_3$/PM loading) were used for analyses of the effects of histamine on Ca$^{2+}$ puffs and global increases in [Ca$^{2+}$]$_c$ (Fig. 3a, Supplementary Fig. 6a). Similar methods, but without EGTA-loading, were used for analyses of global Ca$^{2+}$ signals evoked by histamine (Cal-590) or photolysis of ci-IP$_3$ (Calbryte 590) (Fig. 3l–n); the responses are reported as $\Delta F/F_{max}$, where $\Delta F$ is the peak increase in fluorescence and $F_{max}$ is fluorescence from the Ca$^{2+}$-saturated indicator.

**Measurements of [Ca$^{2+}$]$_c$ in cell populations.** EGFP-IP$_3$R1 HeLa cells were grown in clear-bottomed 96-well plates (Greiner Bio-One, Stonehouse, Gloucester, UK) coated with fibronectin (10 µg ml$^{-1}$) (see the section on Cell culture and transfection). Cells were then washed and loaded with Fluo-8 by incubation in HBS with Fluo-8 AM (2 µM, 60 min, 20 °C), washed and incubated (60 min, 20 °C) before experiments. A FlexStation 3 plate-reader (Molecular Devices) was used to measure Fluo-8 fluorescence at 1.4-s intervals at 20 °C (excitation 490 nm, emission at 525 nm). Responses to histamine were determined in HBS, and responses to ionomycin were determined after the addition of BAPTA in Ca$^{2+}$-free HBS to chelate extracellular Ca$^{2+}$ (final BAPTA concentration = 2.5 mM, free [Ca$^{2+}$] < 40 nM). Fluorescence (F) was recorded (SoftMax Pro, version 5.4, Molecular Devices) and calibrated to [Ca$^{2+}$]$_c$ from [Ca$^{2+}$]$_c = K_D(F − F_{min})/(F_{max} − F)$, where the $K_D$ of Fluo-8 for Ca$^{2+}$ is 389 nM, and $F_{min}$ and $F_{max}$ are the minimal and maximal fluorescence signals determined after addition of Triton X-100 (1%) with either BAPTA (2.5 mM) for $F_{min}$ or CaCl$_2$ (10 mM) for $F_{max}$.

**Quantification of immobile IP$_3$Rs.** For convenience, we often show immobile IP$_3$R puncta by overlaying pseudocoloured images collected at intervals of 30 s, such that the two colours (green and magenta) overlay for immobile IP$_3$Rs (white; e.g., Figs. 2g, h, 4a)[20]. But for all quantitative analyses, we used single-particle tracking (Fiji TrackMate plugin, version 3.8, https://git.io/v6uz2)[73] and trajectory classification (TraJClassifier, version 0.83)[74] to identify immobile IP$_3$R puncta. We confirmed the congruence of the two approaches previously[20]. TrackMate requires selection of unique thresholds for detection of puncta in each image[73], but we confirmed the validity of the thresholds by both manual inspection of images and by demonstrating that the fraction of whole-cell fluorescence attributed to puncta was consistent. For example, in TIRF analyses of cells treated with NS or KRAP siRNA (Fig. 2g, h), the total fluorescence attributed to IP$_3$R puncta relative to whole-cell fluorescence was 56 ± 2% and 53 ± 1%, respectively (mean ± s.e.m., $n$ = 22 cells, $P$ > 0.05, Student's $t$ test). Using TrackMate, the sub-pixel localization of each IP$_3$R punctum in time-lapse TIRF images (usually 10 frames per s for 30 s) was obtained before linking the particles to generate track segments, followed by gap-closing to link the track segments. A displacement threshold of 1.5 µm s$^{-1}$ was employed to track and link particles since the maximal velocity of the particles did not exceed this limit. TraJClassifier, a validated plugin for trajectory classification[74], was used to

classify single-particle trajectories from TrackMate. This algorithm uses a machine-learning approach to classify single-particle trajectories into diffusive, sub-diffusive, confined and directed trajectories. For inclusion in the classification analysis, the minimum trajectory length was 45 frames. This algorithm does not specifically identify immobile particles, which fall within the category of sub-diffusive particles. From each sub-diffusive trajectory, we used the exponent (α) from the plot of mean squared displacement (MSD, $\gamma^2$) versus time ($t$) for anomalous sub-diffusion ($\gamma^2 = 4Dt^\alpha$, where $D$ is the diffusion coefficient)[74,75] and classified particles with α < 0.1 as immobile. The value (α < 0.1) was chosen from the analysis of simulated trajectories performed by Thorsten Wagner (University of Dortmund).

**Statistics.** Most results are presented as mean ± s.d. or s.e.m. from $n$ independent analyses. Statistical comparisons used paired (where indicated) or unpaired, two-tailed Student's $t$ tests, or analysis of variance (ANOVA) with the Bonferroni's correction for multiple comparisons (PRISM version 8, GraphPad, San Diego, CA, USA). Significance levels are shown as *$P$ < 0.05, **$P$ < 0.01, ***$P$ < 0.001 and ****$P$ < 0.0001. Variances of the product of two variables (for Supplementary Fig. 4k) were computed according to ref. [76] (details in Supplementary Table 1). Methods used to assess the statistical significance of colocalization analyses are described in the appropriate methods section. Further statistical details are provided in figure legends and in the summary of all statistical analyses (Supplementary Table 1).

**Reporting summary.** Further information on research design is available in the Nature Research Reporting Summary linked to this article.

## Data availability

All data required to support the conclusions of this paper are provided in Figs. 1–5, Supplementary Figs 1–7, Supplementary Table 1, Supplementary Movies 1–10 and the Source Data File. All software and algorithms used are freely available from sources listed in Methods. Materials and primary data are available from the corresponding authors upon reasonable request. Source data are provided with this paper.

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

## Acknowledgements

The authors thank Martyn Reynolds and Stephen Tovey (Cairn, UK) for help with super-resolution confocal microscopy. Supported by a Wellcome Senior Investigator Award (grant no. 101844), and by a grant (grant no. BB/T012986/1) and studentship (to H.A.S) from the Biotechnology and Biological Sciences Research Council UK. P.A.-A. is a research fellow of Emmanuel College, Cambridge.

## Author contributions

N.B.T. performed most experiments. H.A.S. performed the super-resolution confocal microscopy analyses. P.A.-A. completed analyses of $Ca^{2+}$ signals in cell populations after KRAP knockdown. C.W.T. and N.B.T. conceived the project. C.W.T., N.B.T. P.A.-A. and H.A.S. analysed and interpreted results, and wrote the manuscript.

## Competing interests

The authors declare no competing interests.
