## [Peer Review File · Nature Communications]

REVIEWER COMMENTS

Reviewer #1 (Remarks to the Author):

The present study by Thillaiappan and colleagues documents an important role for KRas-induced actin interacting protein (KRAP) in defining the sub-population of IP3R which are available to respond following agonist stimulation. This is an important contribution because it addresses a puzzling observation that only a relatively small subset of immobile IP3R, close to the plasma membrane respond, while the majority are refractory to stimulation. The Authors demonstrate that clusters of IP3R are tethered to the cortical actin cytoskeleton by an association of a proportion of receptors with KRAP. Further, these privileged IP3R are responsible for Ca²⁺ puffs on stimulation. Indeed, their data, based on siRNA knockdown experiments, suggests that these receptors essentially are the only IP3R capable of opening and thus KRAP expression levels control both the extent of Ca²⁺ release and the localization/trafficking of IP3 to these sites. Given the localization, close to ER-PM junctions, these data may have significant consequences for the gating of store operated Ca²⁺ entry following local depletion.

The paper is well written, the experiments are carefully executed and performed with state-of the art techniques. These factors, in combination with sophisticated analysis make a convincing case for the conclusions presented. My comments mostly relate to the generality of the observations. Further, although this paper is a significant advance and will be of interest to the signaling community, more discussion of previous work performed investigating KRAP and IP3-induced signaling should be included.

I have a few comments.

The Authors suggest that only IP3R associated with KRAP are able to respond and these are highly localized. In some native cell types, such as liver and exocrine cells, Ca²⁺ signals spread from trigger zones (which have been shown to express KRAP/IP3R/Actin) as a wave through the cytoplasm which would seem to occur through IP3R not necessarily associated with KRAP. Some comments on the generality of this phenomenon outside transformed cells and on the other isoforms of IP3R should be made.

Knockdown of KRAP essentially makes Hela cells non-responsive to IP3 and IP3 generating agonists. Although 100 uM Histamine is a maximal concentration in wt Hela's, experiments at higher concentrations in the knockdowns would confirm that KRAP does not simply right- shift the sensitivity of the IP3R to IP3.

In a previous publication, knockdown of KRAP in Heks and MCF7 cells was less effective at reducing agonist stimulated Ca²⁺ signals. These experiments were performed in populations of cells but with apparently similar levels of KRAP reduction. This could of course be a result of heterogeneous expression of siRNA. If I have not missed this, more details should be provided as to how imaging of the siRNA treated cells was performed- were siRNA expressing cells identified prior to imaging, thus negating any impact of heterogeneity on the extent of inhibition?

I am not sure I understand the idea that IP3R are constitutively inhibited by IRBIT etc and that KRAP relieves this inhibition in intact cells (this would seem to be easily accomplished by IP3 itself). The paper might benefit from a further brief mechanistic speculation regarding how KRAP defines the sensitivity of IP3R.

Reviewer #2 (Remarks to the Author):

This paper reveals new insights into the mechanisms regulating calcium puffs - localized subcellular calcium transients that are considered a building block of the calcium signals that regulate numerous functions in virtually all cells of the body. Puffs arise from calcium release at fixed clusters of IP3Rs that are preferentially endowed ('licensed') to respond at low concentrations of IP3, in contrast to the greater number of motile IP3Rs. The mechanisms underlying the immobilization and selective licensing of IP3Rs at puff sites have remained a mystery for several years. This paper represents an important advance by implicating a key molecular player, KRAP. Although this protein had previously been identified as interacting with, and determining the subcellular localization of IP3Rs, the authors here present compelling evidence of superresolution colocalization between KRAP punctae, IP3R clusters and functional puff sites, together with calcium imaging studies relating puff activity to level of KRAP expression. Establishing KRAP as a key player in immobilizing and licensing IP3Rs is an important breakthrough for the calcium signaling field that will open up new experimental avenues and already raises several interesting questions. The novelty and interest of the research thus fully warrant publication in Nature Communications.

I had previously reviewed an earlier version of the manuscript for a different journal. During that review process the authors made extensive revisions that satisfactorily dealt with my original comments. Given that the present manuscript does not appear to be substantially changed or incorporate additional data, I do not think it necessary or appropriate to raise further substantive issues that would require new experiments or time-consuming further revisions. However, there are some points that should be addressed.

I feel the authors do not give appropriate acknowledgement and discussion of prior work. Most importantly, earlier papers already demonstrated that KRAP interacts with IP3Rs and affects their distribution, and that knockdown of KRAP expression impairs calcium liberation through IP3Rs. Although this work is cited (refs 28, 32-35), it is referenced only in a couple of terse sentences, buried in the Results. The authors need to include a more extensive account and introduction to the involvement of KRAP in the Introduction to place their findings in appropriate context. In saying this I do not mean to diminish the importance of the authors findings, which represent a very considerable advance in revealing the interactions and functional affects of KRAP at the levels of individual IP3R clusters and single channels. As another instance, the independence of calcium puffs on extracellular calcium (Fig 1a,b) has previously been reported (cited as ref#11).

The proposal that IP3Rs regulate SOCE through local depletion of ER Ca²⁺ stores is peripheral to the main topic of the paper, and appears an over-interpretation from the data presented (co-localization of KRAP and STIM1 puncta). Also, such a mechanism would seem at variance with the finding that calcium puffs are independent of extracellular calcium.

The suggestion (page 9, top) that IP3Rs in the nuclear envelope (ref #48) may be relieved from constitutive inhibition by KRAP appears in contradiction to the authors own conclusion that only IP3Rs near the plasma membrane require KRAP to be immobilized and licensed.

Reviewer #3 (Remarks to the Author):

Thillaiappan et al.'s article is an extension of their 2017 Nature Communications article in which they showed that Ca⁺ signals are produced by immobile IP3R clusters. Here they hypothesize that the clusters are immobilized to the actin cytoskeleton by the protein KRAP. They support their model by a range of fluorescence microscopy methods that combine super-resolution microscopy and live imaging to investigate calcium signals and show that KRAP immobilizes IP3R clusters to the cytoskeleton, that the cluster size and Ca⁺ signals are constant and global Ca⁺ signal changes with immobilized cluster number.

The article is clearly written and well argued. Most methods have been used in the last article, and the methods largely cite the last article. But there are a number of issues that should be added or

discussed to make the article more accessible to the reader.

The authors never explain how they determined that they have a spatial localization accuracy of 20 nm. I assume the TetraSpeck beads with 100 nm size and very large photon counts cannot be easily used as calibration when imaging is done with standard fluorophores?

If in Fig. 2 the scale bar in the enlarged images is 500 nm, then it looks like the pixel size corresponds to ~ 100 nm. Is this sufficient to obtain a 20 nm localization precision? Again this estimation will depend on the background and the actual photon count rate. But it would be good if it can be shown that this is consistent with the 20 nm localization precision.

In Fig. 1h, the authors show a distribution of center-to-center distances between IP3R and KRAP clusters but explain only that 28% are within 160 nm. But what about the second peak at ~ 500 nm?

The authors have set colocalization cut-offs of 130-160 nm, depending on the technique. The 160 nm is the pixel pitch for the TIRFM. What was the rationale to set the threshold at 160 nm for confocal where the pixel pitch was 65 nm?

To what penetration depth did the authors set the TIRFM? Does this have an influence on the intensity measured by Ca^{2+} puffs?

Reviewer #1

The present study by Thillaiappan and colleagues documents an important role for KRas-induced actin interacting protein (KRAP) in defining the sub-population of IP3R which are available to respond following agonist stimulation. This is an important contribution because it addresses a puzzling observation that only a relatively small subset of immobile IP3R, close to the plasma membrane respond, while the majority are refractory to stimulation. The Authors demonstrate that clusters of IP3R are tethered to the cortical actin cytoskeleton by an association of a proportion of receptors with KRAP. Further, these privileged IP3R are responsible for Ca²⁺ puffs on stimulation. Indeed, their data, based on siRNA knockdown experiments, suggests that these receptors essentially are the only IP3R capable of opening and thus KRAP expression levels control both the extent of Ca²⁺ release and the localization/trafficking of IP3 to these sites. Given the localization, close to ER-PM junctions, these data may have significant consequences for the gating of store operated Ca²⁺ entry following local depletion.

The paper is well written, the experiments are carefully executed and performed with state-of-the-art techniques. These factors, in combination with sophisticated analysis make a convincing case for the conclusions presented. My comments mostly relate to the generality of the observations. Further, although this paper is a significant advance and will be of interest to the signaling community, more discussion of previous work performed investigating KRAP and IP3-induced signaling should be included.

I have a few comments (we have numbered the points to simplify cross-referencing).

- 1. The Authors suggest that only IP₃R associated with KRAP are able to respond and these are highly localized. In some native cell types, such as liver and exocrine cells, Ca²⁺ signals spread from trigger zones (which have been shown to express KRAP/IP₃R/Actin) as a wave through the cytoplasm which would seem to occur through IP₃R not necessarily associated with KRAP. Some comments on the generality of this phenomenon outside transformed cells and on the other isoforms of IP₃R should be made.*

We expanded the description of our previous immunoprecipitation analyses to elaborate our evidence that EGFP-IP₃R1 effectively reports the presence of all three IP₃R subtypes in EGFP-IP₃R1 HeLa cells (p3, para 2). We also added text citing work suggesting that all three IP₃R subtype can interact with KRAP (p4, para 3). The Discussion has been substantially expanded, with additional references, to comment on the likely generality of our findings (p9, para 2) and to discuss implications for the occurrence of Ca²⁺ signals away from the PM, notably in exocrine pancreas (p10, para 1).

- 2. Knockdown of KRAP essentially makes HeLa cells non-responsive to IP₃ and IP₃ generating agonists. Although 100 μM Histamine is a maximal concentration in wt HeLa's, experiments at higher concentrations in the knockdowns would confirm that We repeated the analyses in HeLa cells treated with KRAP siRNA using concentrations of histamine that exceed (by 100-fold) the concentration required for a maximal response in normal cells. The results confirm that loss of responses to histamine in cells without KRAP is not attributable to reduced sensitivity to histamine. The new results are shown in **Fig. 3o** and described on p8, para 1.*

- In a previous publication, knockdown of KRAP in HEKs and MCF7 cells was less effective at reducing agonist stimulated Ca²⁺ signals. These experiments were performed in populations of cells but with apparently similar levels of KRAP reduction. This could of course be a result of heterogeneous expression of siRNA. If I have not missed this, more details should be provided as to how imaging of the siRNA treated cells was performed- were siRNA expressing cells identified prior to imaging, thus negating any impact of heterogeneity on the extent of inhibition?*

In the published analyses of HEK and MCF7 cells¹, siRNA (48 hr) reduced KRAP expression (but it was not quantified) and receptor-evoked Ca²⁺ signals were attenuated, but the extent cannot be quantified because fluorescence was neither calibrated nor shown to avoid saturation. Results with submaximal stimulation (where indicator saturation did not occur) suggest ~50% inhibition of Ca²⁺ signals after KRAP knockdown (Fig. 1B and 2A in¹).

In our single-cell analyses of Ca²⁺ signals after treatment with KRAP-siRNA (72 hr), cells were randomly selected with no attempt to exclude untransfected cells. Our WB indicates ~75% knockdown of KRAP expression with siRNA (**Supplementary Fig. 4b**), suggesting at least 75% transfection efficiency. As noted in **Fig. S4g**, we rarely found siRNA-treated cells with residual KRAP. We have now quantified the fraction of cells in which there is detectable KRAP after siRNA treatment. The results establish that after siRNA treatment, KRAP is undetectable in 90% of cells (p14, para 2). We conclude that most cells treated with KRAP siRNA have much reduced expression of KRAP, consistent with loss of Ca²⁺ signals recorded from single cells (**Fig. 3l-n**) and cell populations (**Fig. 3o**, p14, para 2). We expanded the Methods (*Treatment with siRNA*) to explain both the analysis of KRAP expression and random selection of cells for analysis (p14, para 2).

- I am not sure I understand the idea that IP₃R are constitutively inhibited by IRBIT etc and that KRAP relieves this inhibition in intact cells (this would seem to be easily accomplished by IP₃ itself). The paper might benefit from a further brief mechanistic speculation regarding how KRAP defines the sensitivity of IP₃R.*

Our concern relates to the observation that supramaximal concentrations of IP₃ or histamine (see #2) fail to evoke Ca²⁺ release in intact cells without KRAP (indicating an obligatory need for KRAP), yet isolated IP₃Rs open after IP₃ addition (suggesting that in these 'disrupted' systems KRAP is not essential). We speculate that IP₃Rs in intact cells are tonically inhibited and that KRAP relieves the inhibition. We expanded the Discussion (p9-10) to more clearly elaborate these speculations.

Reviewer #2

This paper reveals new insights into the mechanisms regulating calcium puffs - localized subcellular calcium transients that are considered a building block of the calcium signals that regulate numerous functions in virtually all cells of the body. Puffs arise from calcium release at fixed clusters of IP₃Rs that are preferentially endowed ('licensed') to respond at low concentrations of IP₃, in contrast to the greater number of motile IP₃Rs. The mechanisms underlying the immobilization and selective licensing of IP₃Rs at puff sites have remained a mystery for several years. This paper represents an important advance by implicating a key molecular player, KRAP. Although this protein had previously been identified as interacting with, and determining the subcellular localization of IP₃Rs, the authors here present compelling evidence of superresolution colocalization between KRAP

punctae, IP₃R clusters and functional puff sites, together with calcium imaging studies relating puff activity to level of KRAP expression. Establishing KRAP as a key player in immobilizing and licensing IP₃Rs is an important breakthrough for the calcium signaling field that will open up new experimental avenues and already raises several interesting questions. The novelty and interest of the research thus fully warrant publication in Nature Communications.

I had previously reviewed an earlier version of the manuscript for a different journal. During that review process the authors made extensive revisions that satisfactorily dealt with my original comments. Given that the present manuscript does not appear to be substantially changed or incorporate additional data, I do not think it necessary or appropriate to raise further substantive issues that would require new experiments or time-consuming further revisions. However, there are some points that should be addressed (we have numbered the points to simplify cross-referencing).

- 1. I feel the authors do not give appropriate acknowledgement and discussion of prior work. Most importantly, earlier papers already demonstrated that KRAP interacts with IP₃Rs and affects their distribution, and that knockdown of KRAP expression impairs calcium liberation through IP₃Rs. Although this work is cited (refs 28, 32-35), it is referenced only in a couple of terse sentences, buried in the Results. The authors need to include a more extensive account and introduction to the involvement of KRAP in the Introduction to place their findings in appropriate context. In saying this I do not mean to diminish the importance of the authors findings, which represent a very considerable advance in revealing the interactions and functional affects of KRAP at the levels of individual IP₃R clusters and single channels. As another instance, the independence of calcium puffs on extracellular calcium (Fig 1a,b) has previously been reported (cited as ref#11).*

We accept that in trying to be concise, we had been ungenerous in our description of preceding work. We revised the text to provide much fuller descriptions of previous work linking KRAP and IP₃Rs (p4, para 3; and additional references), and to recognize previous reports that Ca²⁺ puffs can occur in the absence of extracellular Ca²⁺ (p3, para 3).

- 2. The proposal that IP₃Rs regulate SOCE through local depletion of ER Ca²⁺ stores is peripheral to the main topic of the paper, and appears an over-interpretation from the data presented (co-localization of KRAP and STIM1 puncta). Also, such a mechanism would seem at variance with the finding that calcium puffs are independent of extracellular calcium.*

The Ca²⁺-sensitivity of STIM1 suggests that its activation should require substantial loss of ER Ca²⁺, yet if that occurred throughout the ER it might compromise other ER functions. Our proposal that licensed IP₃Rs mediate local and substantial depletion of ER Ca²⁺ alongside sites where STIM1 accumulates provides a possible resolution to this conundrum. The proposal is speculative, but it rests on our evidence that STIM1 accumulates alongside licensed IP₃Rs. We suggest that it is appropriate, in the final section of the Discussion and with appropriate caution, to speculate on the possible role of licensed IP₃Rs in regulating SOCE. We modified the text (p10, para 2) to ensure that the speculative nature of our proposal is clear.

Local regulation of SOCE by licensed IP₃Rs need not imply that Ca²⁺ puffs require Ca²⁺ entry. We have (#1) reinforced the text stating that Ca²⁺ puffs are unaffected by removal of extracellular Ca²⁺.

3. *The suggestion (page 9, top) that IP₃Rs in the nuclear envelope (ref #48) may be relieved from constitutive inhibition by KRAP appears in contradiction to the authors own conclusion that only IP₃Rs near the plasma membrane require KRAP to be immobilized and licensed.*

We expanded the Discussion to suggest explanations for the opening of IP₃Rs in disrupted systems devoid of KRAP (p9, para 2) (Reviewer 1, #4) and to suggest possible mechanisms for opening of IP₃Rs remote from the PM in some cells (p10, para 1) (Reviewer 1, #1).

Reviewer #3

Thillaiappan et al.'s article is an extension of their 2017 Nature Communications article in which they showed that Ca⁺ signals are produced by immobile IP₃R clusters. Here they hypothesize that the clusters are immobilized to the actin cytoskeleton by the protein KRAP. They support their model by a range of fluorescence microscopy methods that combine super-resolution microscopy and live imaging to investigate calcium signals and show that KRAP immobilizes IP₃R clusters to the cytoskeleton, that the cluster size and Ca⁺ signals are constant and global Ca⁺ signal changes with immobilized cluster number.

The article is clearly written and well argued. Most methods have been used in the last article, and the methods largely cite the last article. But there are a number of issues that should be added or discussed to make the article more accessible to the reader (we have numbered the points to simplify cross-referencing).

1. *The authors never explain how they determined that they have a spatial localization accuracy of 20 nm. I assume the TetraSpeck beads with 100 nm size and very large photon counts cannot be easily used as calibration when imaging is done with standard fluorophores?*

We repeated our estimation of the STORM resolution² using small objects in the STORM images generated with AlexaFluor 647. The results suggest a resolution (FWHM) of 24.2 ± 5.6 nm, comparable to that determined using TetraSpeck beads (21.8 ± 4 nm). The Methods (STORM, p17, para 3) and legend to **Fig. 2a,b**.

2. *If in Fig. 2 the scale bar in the enlarged images is 500 nm, then it looks like the pixel size corresponds to ~ 100 nm. Is this sufficient to obtain a 20 nm localization precision? Again this estimation will depend on the background and the actual photon count rate. But it would be good if it can be shown that this is consistent with the 20 nm localization precision.*

The areas shown in the enlarged areas (overlays of STORM and TIRF) show pixelation from the TIRF images (pixel size 100 nm), but they are too large to reveal pixelation within the STORM image (pixel size 10 nm, ie 100 pixels within each of the evident pixels). The localization precision of ~20-25 nm (see #1) relates only to the STORM image. We amended the legend to **Fig. 2a,b** to avoid confusion.

3. *In Fig. 1h, the authors show a distribution of center-to-center distances between IP₃R and KRAP clusters but explain only that 28% are within 160 nm. But what about the second peak at ~500 nm?*

The densities of IP₃R and KRAP puncta are such that when randomly distributed, the mean centre-centre distance from an IP₃R to the nearest KRAP is 450 nm (14886 puncta from 22 cells). This accounts for what the reviewer calls a 'second peak'. We amended **Fig. 1h** to show both observed and randomized distance distributions.

4. *The authors have set colocalization cut-offs of 130-160 nm, depending on the technique. The 160 nm is the pixel pitch for the TIRFM. What was the rationale to set the threshold at 160 nm for confocal where the pixel pitch was 65 nm?*

We aimed to keep colocalization criteria as similar as possible between TIRFM (160 nm, 1 pixel) and superresolution confocal microscopy (130 nm (*not* 160 nm), 2 pixels). It would be inappropriate to use a single pixel for the latter because the theoretical lateral resolution is ~120 nm. We expanded the Methods (*Colocalization analyses*, p18, para 2) to explain the criteria.

5. *To what penetration depth did the authors set the TIRFM? Does this have an influence of the intensity measured by Ca²⁺ puffs?*

We are a little puzzled by this comment. The penetration depth for TIRFM (90-120 nm) is reported in Methods (p16, para 1). The only data reporting the intensity (ie amplitude) of Ca²⁺ puffs is shown in **Fig. 4e,f**, where the depth was the same for the two conditions.

References

¹Fujimoto, T et al (2011) *Biochem Biophys Res Commun* **408**, 214-7.

²Demmerle, J et al (2015) *Methods* **88**, 3-10.

REVIEWERS' COMMENTS

Reviewer #1 (Remarks to the Author):

The Authors have satisfactorily answered all my concerns. Thank you for giving me the opportunity to read this important and interesting paper.

Reviewer #2 (Remarks to the Author):

The authors have satisfactorily responded to my comments. I believe that this interesting and important paper is now acceptable for publication.

Reviewer #3 (Remarks to the Author):

The authors have answered my questions and I have no further questions or comments.